# *lnc-IL7R* Expression Reflects Physiological Pulmonary Function and Its Aberration Is a Putative Indicator of COPD

**DOI:** 10.3390/biomedicines10040786

**Published:** 2022-03-28

**Authors:** Oluwaseun Adebayo Bamodu, Sheng-Ming Wu, Po-Hao Feng, Wei-Lun Sun, Cheng-Wei Lin, Hsiao-Chi Chuang, Shu-Chuan Ho, Kuan-Yuan Chen, Tzu-Tao Chen, Chien-Hua Tseng, Wen-Te Liu, Kang-Yun Lee

**Affiliations:** 1Division of Pulmonary Medicine, Department of Internal Medicine, Shuang Ho Hospital, Taipei Medical University, New Taipei City 235, Taiwan; dr_bamodu@yahoo.com (O.A.B.); chitosan@tmu.edu.tw (S.-M.W.); fengpohao@tmu.edu.tw (P.-H.F.); brontesun@gmail.com (W.-L.S.); chuanghc@tmu.edu.tw (H.-C.C.); shu-chuan@tmu.edu.tw (S.-C.H.); 14388@s.tmu.edu.tw (K.-Y.C.); 09330@s.tmu.edu.tw (T.-T.C.); chtseng0925@tmu.edu.tw (C.-H.T.); 2Department of Medical Research, Shuang Ho Hospital, Taipei Medical University, New Taipei City 235, Taiwan; 3Department of Urology, Shuang Ho Hospital, Taipei Medical University, New Taipei City 235, Taiwan; 4Division of Hematology and Oncology, Department of Internal Medicine, Shuang Ho Hospital, Taipei Medical University, New Taipei City 235, Taiwan; 5Division of Pulmonary Medicine, Department of Internal Medicine, School of Medicine, College of Medicine, Taipei Medical University, Taipei 110, Taiwan; 6Division of Clinical Care Medicine, Department of Emergency and Critical Care Medicine, Shuang Ho Hospital, Taipei Medical University, New Taipei City 235, Taiwan; 7TMU Research Center of Thoracic Medicine, Taipei Medical University, Taipei 110, Taiwan; cwlin@tmu.edu.tw; 8Department of Biochemistry and Molecular Cell Biology, School of Medicine, College of Medicine, Taipei Medical University, Taipei 110, Taiwan; 9International PhD Program for Cell Therapy and Regeneration Medicine, College of Medicine, Taipei Medical University, Taipei 110, Taiwan; 10School of Respiratory Therapy, College of Medicine, Taipei Medical University, Taipei 110, Taiwan; 11Graduate Institute of Clinical Medicine, College of Medicine, Taipei Medical University, Taipei 110, Taiwan; 12Institute of Epidemiology and Preventive Medicine, College of Public Health, National Taiwan University, Taipei 106, Taiwan

**Keywords:** pulmonary function, chronic obstructive pulmonary disease (COPD), lung inflammation, long noncoding RNA, *lnc-IL7R*, GOLD stage, %LAA_-950insp_, FEV_1_(%), FVC(%)

## Abstract

Despite rapidly evolving pathobiological mechanistic demystification, coupled with advances in diagnostic and therapeutic modalities, chronic obstructive pulmonary disease (COPD) remains a major healthcare and clinical challenge, globally. Further compounded by the dearth of available curative anti-COPD therapy, it is posited that this challenge may not be dissociated from the current lack of actionable COPD pathognomonic molecular biomarkers. There is accruing evidence of the involvement of protracted ‘smoldering’ inflammation, repeated lung injury, and accelerated lung aging in enhanced predisposition to or progression of COPD. The relatively novel uncharacterized human long noncoding RNA *lnc-IL7R* (otherwise called LOC100506406) is increasingly designated a negative modulator of inflammation and regulator of cellular stress responses; however, its role in pulmonary physiology and COPD pathogenesis remains largely unclear and underexplored. Our previous work suggested that upregulated *lnc-IL7R* expression attenuates inflammation following the activation of the toll-like receptor (TLR)-dependent innate immune system, and that the upregulated *lnc-IL7R* is anti-correlated with concomitant high PM_2.5_, PM_10_, and SO_2_ levels, which is pathognomonic for exacerbated/aggravated COPD in Taiwan. In the present study, our quantitative analysis of *lnc-IL7R* expression in our COPD cohort (n = 125) showed that the *lnc-IL7R* level was significantly correlated with physiological pulmonary function and exhibited COPD-based stratification implications (area under the curve, AUC = 0.86, *p* < 0.001). We found that the *lnc-IL7R* level correctly identified patients with COPD (sensitivity = 0.83, specificity = 0.83), precisely discriminated those without emphysematous phenotype (sensitivity = 0.48, specificity = 0.89), and its differential expression reflected disease course based on its correlation with the COPD GOLD stage (r = −0.59, *p* < 0.001), %LAA_-950insp_ (r = −0.30, *p* = 0.002), total LAA (r = −0.35, *p* < 0.001), FEV_1_(%) (r = 0.52, *p* < 0.001), FVC (%) (r = 0.45, *p* < 0.001), and post-bronchodilator FEV_1_/FVC (r = 0.41, *p* < 0.001). Consistent with other data, our bioinformatics-aided dose–response plot showed that the probability of COPD decreased as *lnc-IL7R* expression increased, thus, corroborating our posited anti-COPD therapeutic potential of *lnc-IL7R*. In conclusion, reduced *lnc-IL7R* expression not only is associated with inflammation in the airway epithelial cells but is indicative of impaired pulmonary function, pathognomonic of COPD, and predictive of an exacerbated/ aggravated COPD phenotype. These data provide new mechanistic insights into the ailing lung and COPD progression, as well as suggest a novel actionable molecular factor that may be exploited as an efficacious therapeutic strategy in patients with COPD.

## 1. Clinical Relevance

(i)Protracted ‘smoldering’ inflammation, repeated lung injury, and accelerated lung aging are implicated in the enhanced predisposition to or progression of COPD;(ii)Altered *lnc-IL7R* expression in the human airway epithelium and blood is clinically relevant in pulmonary pathophysiology;(iii)The *lnc-IL7R* level is significantly downregulated in patients with COPD and is correlated with disease progression;(iv)The correlation of circulating *lnc-IL7R* with improved pulmonary function, and anti-correlation with PM_2.5_, PM_10_, and SO_2_ provide some mechanistic insights into pulmonary dysfunction and COPD, as well as suggest potential actionable novel biotherapeutics/biologics for the treatment of COPD.

## 2. Introduction

Chronic obstructive pulmonary disease (COPD), a noncommunicable disease group characterized by progressive but preventable airflow obstruction and impaired respiration, including chronic bronchitis, refractory asthma, and emphysema, constitutes a major global health challenge [1,2]. Based on the World Health Organization’s Global Health Estimates, accounting for approximately 100,000,000 lost healthy life-years in 2019 alone, compared with 2000, COPD now ranks as the third leading cause of mortality globally [2]. 

As already demonstrated by our team and others, long-term exposure to air pollutants, including nitrogen oxides (NO_x_), ambient ozone (O_3_), emitted hydrocarbons (HC), and fine particulate matter <10 μm (PM_10_) or <2.5 μm (PM_2.5_) in aerodynamic diameter, is implicated in the progressive decline in pulmonary function in patients with COPD/emphysema and cannot be decoupled from reported reduction in lung function indices, including the forced vital capacity (FVC), forced expiratory volume in 1 sec (FEV_1_), maximum mid-expiratory flow (MMEF), and FEV_1_/FVC ratio elicited by the cumulative increase in ambient air pollution [3].

Primarily affecting the lungs, there is increasing evidence implicating chronic systemic inflammation in the progressive largely irreversible airflow obstruction that is characteristic of COPD [4,5]. This chronic inflammation is driven by a cascade of both nonspecific innate and specific acquired immune responses in the lungs, and though more predominant in the lung parenchyma and bronchial walls of the small airways [5], differences in the nature of such inflammation, and its location define the COPD phenotype, namely chronic bronchitis, refractory asthma, or emphysema, as well as influence clinical course and therapy response [5,6]. The flaring of this COPD-associated inflammation during acute exacerbation or aggravation of COPD is characterized by increased tissue pooling of alveolar macrophages, neutrophils, Tc1, Th1, and Th17 lymphocytes, as well as innate lymphoid cells (ILCs) from the circulation [6,7]. In concert with structural cells of the epithelium, endothelium, and fibroblasts, these pooled cells secrete various mediators of inflammation, including chemokines, cytokines, lipid mediators, and growth factors [7].

Systemic inflammation is increasingly implicated in accelerated lung aging and associated decrease in lung function [8]. Several studies indicate that one of every two patients with COPD exhibit accelerated decrease in pulmonary function [8], which, unlike ‘senile emphysema’ that is associated with the physiological aging of the lungs with resultant enlarged alveolar spaces and loss of lung elasticity in the elderly, is conversely defined by marked alveolar wall destruction and peripheral airway fibrosis [8,9]. This lung injury can result in permanent reduction in the gas exchange surface area and respiratory function of patients with COPD [10]. The loss of pulmonary elasticity due to the proteolytic destruction of lung parenchyma and alveolar walls plays a major role in COPD pathognomonic airway obstruction, and while the irreversibility of such lung damage regardless of pharmacologic therapy has been opined [8,9], we posit that with improved understanding of COPD pathobiology and further unraveling of its underlying mechanism, not only is reduced disease progression through inhibition of proinflammation and associated enzymatic signaling possible, but lung tissue regeneration and reversibility of obstructive defects, with consequent improved pulmonary function, is probable [10,11,12].

Aside from innate immune cells, the epitranscriptomic memory potential of lung epithelium has been suggested to drive immune responses associated with mucus hyperreactivity and remodeling of the airway related to COPD. While several susceptibility genes have been implicated in airway remodeling, the culpability of long noncoding RNAs (lncRNAs), such as *lnc-IL7R* is increasingly documented in the regulation of postexposure airway inflammation, COPD pathogenesis, and disease progression [3,13,14].

The present study, exploring for causative association between *lnc-IL7R* and COPD, provides evidence that the elevated profile of circulating *lnc-IL7R* reflects improved pulmonary function, and is anti-correlated with effectors of pulmonary dysfunction PM_2.5_, PM_10_, and SO_2_, suggesting the therapeutic feasibility of *lnc-IL7R* as a potential actionable novel biotherapeutic/biologic for the treatment of COPD.

## 3. Methods

In the present single-center, prospective, non-randomized study, we enrolled 125 patients with COPD, diagnosed consistent with international recommendations, as described in our previous publication [3]. Patients were clinically stable (without exacerbation, at least for one month), without significant comorbidities or history of systemic corticosteroids, diuretics, cytotoxic agents, or alcohol abuse (alcohol/day > 40 g/day). COPD severity was based on the Global Initiative for Chronic Obstructive Lung Disease (GOLD) guidelines [15], and the percentages of low attenuation area below −950 Hounsfield units (%LAA_-950insp_) defined classification into no/mild emphysema (%LAA_-950insp_ < 6), moderate emphysema (6 ≤ %LAA_-950insp_ < 14), and severe emphysema (%LAA_-950insp_ ≥ 14) [3,16]. Patients were either current smokers (n = 49), ex-smokers (n = 66), or never smokers (n = 10). Hematologic variables, including blood *lnc-IL7R*, were derived from routine blood workup. Other data retrieved included smoking history (status, pack-year), anthropometric (height, weight, body mass index), geospatial (location coordinates), ambient air pollution, spirometric (FEV_1_, FVC, FEV_1_/FVC), and computed tomography (%LAA_-950insp_, total LAA) variables. Oxygen saturation (SvO_2_, SpO_2_) levels were measured using a Sibelmed^®^ Datospir micro C portable spirometer (Sibel, S.A.U., Barcelona, Spain). Bronchodilator response (BDR) was calculated using the formula:BDR=[(FEV1post−BDFEV1pre−BD)−1]×100%
where change in FEV_1_ ≥ 12% and 400 mL defined a positive BDR. Clinical sample preparation and quantitative reverse transcription PCR (RT-qPCR) for determination of the *lnc-IL7R* level in the whole blood of participants were strictly performed as earlier described [3].

### Statistical Analysis

All data are expressed as mean ± standard deviation (SD) or percentages (%). Differences between categorical variables was compared using Pearson’s chi-squared (χ^2^) test. The paired *t*-test was used for comparing two continuous variables. The Student’s *t*-test was used to assess alterations in pulmonary function based on %LAA_-950insp_, FEV_1_, FVC, FEV_1_/FVC, and *lnc-IL7R* expression levels. Comparison of ≥3 categorical variables was performed using the Kruskal-Wallis test with post hoc Dunn’s multiple comparison test. Spearman’s rank correlation was used to determine the relationship between variables. *p*-value ≤ 0.05 defined statistical significance. All statistical analyses were performed using IBM SPSS Statistics for Windows, Version 25.0 (IBM Corp. Released 2017, Armonk, NY, USA: IBM Corp). Geospatial visualization and analysis were performed using the ArcGIS server software version 10.8.1 (ESRI, Redlands, CA, USA).

## 4. Results

### 4.1. Baseline Characteristics of Our COPD Cohort

The baseline characteristics of our COPD cohort (n = 125 vs. calculated required sample size of 89) are presented in Table 1. According to the COPD GOLD guidelines (15), 14.4% of all COPD patients exhibited mild COPD (FEV_1_ = 85.4 ± 5.2%), while 46.4%, 30.4%, and 8.8% were moderate (FEV_1_ = 63.8 ± 8.6%), severe (FEV_1_ = 40.0 ± 5.7%), and very severe (FEV_1_ = 24.8 ± 4.0%) COPD cases, respectively (Table 1). Across all GOLD stage groups, the intergroup differences were statistically significant. All pulmonary function indices, namely FEV_1_(L), FEV_1_(%), and FEV_1_/FVC(%), were inversely correlated with smoking pack-years and COPD exacerbation in the previous year but positively correlated with *lnc-IL7R* expression (Table 1). Compared with patients with mild COPD (GOLD I), 29%, 59%, or 70% reduced median *lnc-IL7R* expression was observed in patients with moderate (GOLD II), severe (GOLD III), or very severe (GOLD IV) COPD, respectively (Table 1, also see Appendix A). Females were more predisposed to severe COPD (GOLD stage III/IV) than their male counterparts (69.2% female vs. 35.7% male) (Table 1).

The mean FEV_1_ and FVC of our patients were 56.3 ± 19.2% and 79.80 ± 18.47%, respectively (Appendix A). 

### 4.2. lnc-IL7R Expression Is Associated with Physiological Pulmonary Function and Exhibits Diagnostic Relevance for COPD-Based Patient Stratification

Corroborating the results above, principal component analyses showed that while smoking history, pack-year, age, total LAA, and %LAA_-950insp_ (component 2) were anti-correlated with normal lung function, male sex, BMI, FEV_1_, FVC, FEV_1_/FVC, and *lnc-IL7R* (component 1) were positively correlated with same (Figure 1A). In concordance, component 1 factors were negatively correlated with GOLD COPD severity, while component 2 variables positively correlated with it (Figure 1B). Moreover, associative ellipsoid 3D visualization showed that increasing *lnc-IL7R* expression profile was associated with ameliorating COPD and %LAA_-950insp_-based emphysema severity (Figure 1C). Of great clinical relevance, *lnc-IL7R* expression exhibited excellent capability to discriminate between patients with no/mild COPD (GOLD I) and exacerbated/aggravated cases (GOLD II, III, IV) (area under the curve, AUC = 0.86, *p* < 0.001) (Figure 1D,E). A Youden’s J index of 0.66 (sensitivity = 83.0, specificity = 83.3) affirmed *lnc-IL7R* diagnostic relevance, with an optimal threshold value/cutoff point ≤ 0.64 defining exacerbated COPD (Figure 1E).

### 4.3. lnc-IL7R Level Correlates with COPD Status and Emphysematous Phenotype, and Its Differential Expression Reflects Disease Course

Consistent with Figure 1E, we observed that subjects with *lnc-IL7R* expression ≤0.64 were invariably COPD cases (Figure 2A), and cases with emphysematous phenotype were defined by an *lnc-IL7R* expression ≤0.46 (Figure 2B,C). Correlative analyses further demonstrated that *lnc-IL7R* expression was anti-correlated with COPD severity (r = 0.59, *p* < 0.001), emphysema severity (r = 0.30, *p* = 0.002), and emphysema status (r = 0.35, *p* < 0.001) (Figure 2D–F). Conversely, *lnc-IL7R* expression was positively correlated with pulmonary function indices, namely FEV_1_ (r = 0.52, *p* < 0.001), FVC (r = 0.45, *p* < 0.001), and the post-bronchodilator (post-BD) FEV_1_/FVC ratio (r = 0.41, *p* < 0.001).

### 4.4. lnc-IL7R, a Probable Biological Response Modifier, Exhibits Strong Anti-COPD Therapeutic Potential

Furthermore, using bioinformatics-aided dose–response modeling we simulated the relationship between the internal dose metric, namely *lnc-IL7R* expression, and susceptibility to COPD. We observed that as the expression level of *lnc-IL7R* increased, the probability of COPD decreased with a threshold value of ~0.65 (Chi-squared, *X*^2^ = 11.95, *p* = 0.0005) (Figure 3A), which is reminiscent of a pharmacological effective dose (ED), where the ED is the concentration of *lnc-IL7R* that elicits a strong anti-COPD biological response. Consistent with this, after numeralization of the GOLD stage (0, GOLD 1; 1, GOLD II; 2, GOLD III; 3, GOLD IV), we showed that while high and moderate expression of *lnc-IL7R* is associated with no/mild COPD (GOLD I) and moderate COPD (GOLD II), patients with severe cases (GOLD III, IV) were *lnc-IL7R* negative (Figure 3B). Similarly, compared with the *lnc-IL7R* nonexpressors with severe emphysema, *lnc-IL7R* expressors, dependent on expression level, exhibited no, mild, or moderate emphysema (Figure 3C). Consistent with these results, we also found a statistically significant linear correlation between *lnc-IL7R* expression, post-BD FEV_1_/FVC ratio, and predictor FEV_1_ (%) (Figure 3D). In corroboration, understanding that bronchodilator response (BDR) reflects eosinophilic airway inflammation, bronchial atrophy and hyperactivity, lung function decline, and is a major criterion for the diagnosis of the asthma-COPD overlap syndrome, as well as a potential marker for delineating different types of COPD phenotypes [17,18], added to the statistically significant correlation between *lnc-IL7R* expression and post-BD FEV_1_/FVC ratio (r = 0.41, *p* < 0.001) (Appendix A), we observed higher *lnc-IL7R* levels in patients with negative BDR (FEV_1_ < 12% and 400 mL), compared with their BDR positive peers (1.5-fold, *p* < 0.001) (Appendix A). More so, the concomitant increase in *lnc-IL7R* levels and post-BD FEV_1_/FVC ratio was mostly associated with BDR negative status (coefficient of determination, R^2^ = 0.17; F-ratio = 17.62; *p* = 0.0001) (Appendix A). These data indicate that higher *lnc-IL7R* reflects an absence of airflow obstruction, is a probable biological response modifier, and exhibits strong anti-COPD therapeutic potential. 

## 5. Discussion

Consistent with contemporary knowledge that only a dismal proportion (<1%) of all proposed biomarkers end up being translated to clinical utility [19,20], the identification of disease pathognomonic biomarkers that inform efficacious disease management or facilitate novel therapeutic approaches continues to lag behind the significant increase in our understanding of COPD pathophysiology and pathogenesis in the past decades. Against this background, and accentuating the unwritten consensus in pulmonary medicine that “no biomarker other than lung function has been shown to be useful, to date, for the diagnosis of COPD” [21], the present study, for the first time to the best of our knowledge, demonstrates that *lnc-IL7R* expression level (i) is associated with physiological pulmonary function, (ii) correlates with COPD status and emphysematous phenotype, and (iii) exhibits diagnostic relevance for COPD-based patient stratification. We also demonstrated that (iv) *lnc-IL7R* differential expression reflects disease course, and that (v) *lnc-IL7R*, a probable biological response modifier, exhibits strong anti-COPD therapeutic potential.

The strong positive correlation between *lnc-IL7R* expression levels and spirometric pulmonary function indices, exemplified by a 4.9-fold reduction in *lnc-IL7R* expression level eliciting marked decline in FEV_1_ (L) (3.3-fold, r = 0.37), FEV_1_ (%) (3.4-fold, r = 0.52), and post-bronchodilator FEV_1_/FVC ratio (1.5-fold, r = 0.41) is of clinical significance especially as the contemporary definition of COPD is based on a fixed FEV_1_/FVC ratio or on the lower limits of FEV_1_/FVC of a healthy reference population [22]. Furthermore, cognizant of an evolving understanding that the presence of reduced post-bronchodilator FEV1/FVC (usually <0.70 per GOLD criteria) may not necessarily be the strongest confirmation of the presence of persistent airflow limitation or predictor of COPD progression and eventual disease-specific mortality [22,23], the present study also provided clinical evidence that *lnc-IL7R* expression level was not only anti-correlated with spirometric GOLD-based COPD severity (r = −0.59) but also with computer tomography-based total LAA (%) (r = −0.35), %LAA_-950insp_ (r = −0.30), and history of clinical COPD exacerbation in previous year. This is particularly interesting, as it demonstrates that unlike its spirometric predecessors, our proposed *lnc-IL7R* expression-based diagnosis and/or prognosis of COPD takes into account COPD-pathognomonic lung inflammation and tissue damage that are captured with CT imaging, thus connoting a broader definition of abnormal lung function, which is purportedly missed by the spirometric definition of COPD in ~40% of the population, even in the presence of smoking history and displayed COPD symptomatology [24].

Our present findings also lend some credence to the recently proposed dual axes pathophysiological basis of COPD, namely, airway-predominant and emphysema-predominant [24], especially with *lnc-IL7R* expression serving as a probable molecular bridge between the suppressed spirometric PFT indices (airway-predominant component) and increased CT imaging indices reflective of emphysema status and severity (emphysema-predominant). Consistent with the fundamental principle of GOLD that any clinically feasible COPD diagnostics should be simple and applicable worldwide [15], and in light of the seemingly impracticability of the COPDGene’s proposal, which is largely dependent on the relatively expensive CT imaging modality, in low-income countries [15,24], considering its ability to recapitulate CT evidence of airway inflammation and/or emphysema [3,14], the present study proposes the clinical utility of *lnc-IL7R* expression as a surrogate biomarker of COPD based on its pathognomonic attribution as demonstrated herein. In terms of early identification and initiation of appropriate therapy, we posit that the altered expression of *lnc-IL7R* may be diagnostically reliable in that group of patients who may fall outside the current spirometric GOLD COPD definition but are indeed at risk of COPD, actually at high risk of COPD-specific death, or exhibit COPD-like symptoms.

Our data, which for the first time, to the best of our knowledge, show that as the expression level of *lnc-IL7R* increased, the probability of COPD decreased profoundly, with associated statistically significant linear correlation between *lnc-IL7R* expression, post-BD FEV_1_/FVC ratio, and predicted FEV_1_ (%), is diagnostically relevant. The diagnosis of COPD, as per the GOLD guidelines [15,16], requires post-bronchodilator FEV_1_/FVC < 0.70 combined with FEV_1_ < 0.80 of the predicted value, thus, highlighting the clinical significance of the demonstrated concomitant increase in *lnc-IL7R* expression and the spirometric indicators of pulmonary function and/or health. In concert with other findings documented herein, we posit that altered *lnc-IL7R* expression may be a diagnostically valuable indicator of altered pulmonary function, and a clinically feasible, readily accessible, and relatively cheap surrogate biomarker of preventable and treatable COPD, which is usually insidious and associated with an aberrant inflammatory response of the lungs to noxious particulate matters or gases.

In conclusion, the present study provides some evidence that reduced *lnc-IL7R* expression is associated with inflammation in the airway epithelial cells, indicative of impaired pulmonary function, pathognomonic of COPD and predictive of an exacerbated/ aggravated COPD phenotype. These data provide new mechanistic insight into the ailing lung and COPD progression, as well as suggest a novel actionable molecular factor that may be exploited as an efficacious therapeutic strategy in patients with COPD.

## Figures and Tables

**Figure 1 biomedicines-10-00786-f001:**
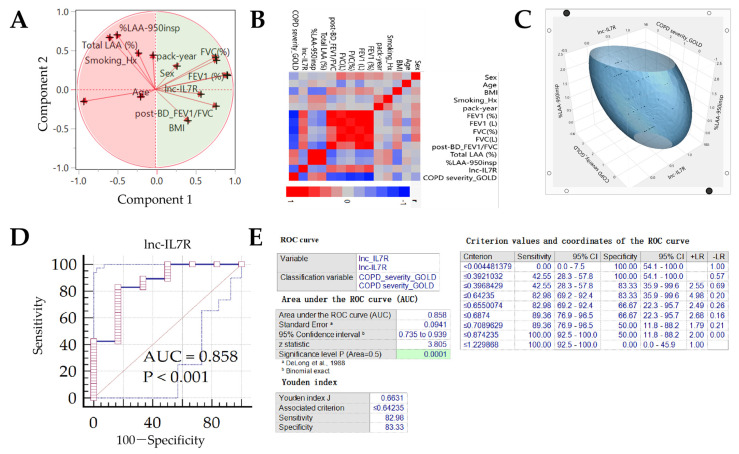
*lnc-IL7R* positively correlates with physiological pulmonary function and exhibits diagnostic relevance for COPD-based patient stratification. (**A**) Visualization of principal component analysis based on factor analysis loading showing the associative predisposition of our panel of COPD-related variables and their stratification into components 1 and 2. (**B**) Correlative heat map of our panel of COPD-related variables. (**C**) Ellipsoid 3D image of the association between *lnc-IL7R*, GOLD-defined COPD severity, and %LAA_-950insp_-based Emphysema severity. (**D**) Graphical depiction of the ROC curve and AUC value of *lnc-IL7R* expression in our COPD cohort. (**E**) Statistical chart of the ROC curve of *lnc-IL7R* expression. ROC, receiver operating characteristic; AUC, area under curve.

**Figure 2 biomedicines-10-00786-f002:**
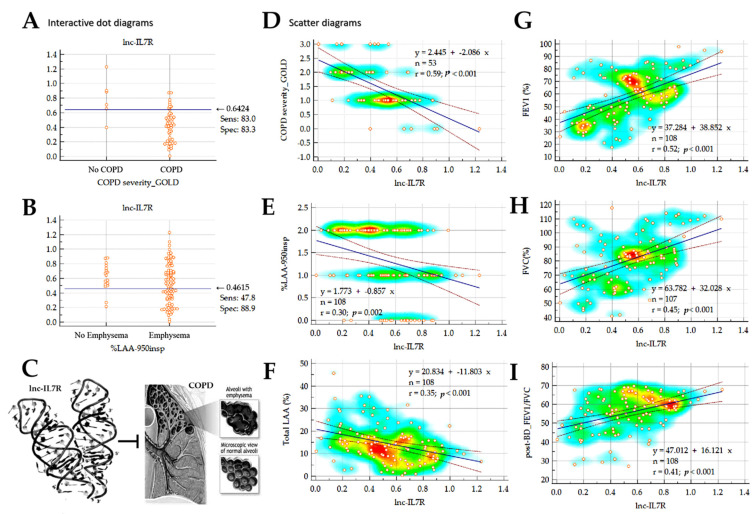
*lnc-IL7R* level correlates with COPD status and emphysematous phenotype, and its differential expression reflects disease course. Dot plots showing stratification of patients into (**A**) No-COPD or COPD, and (**B**) No-Emphysema or Emphysema groups, based on the expression of *lnc-IL7R*. (**C**) Pictorial depiction of the inhibitory role of *lnc-IL7R* expression on COPD with/or emphysematous phenotype. Scatter heat map plots depicting the correlation between *lnc-IL7R* expression, and (**D**) GOLD-defined COPD severity, (**E**) %LAA_-950insp_-based Emphysema severity, (**F**) total LAA, (**G**) FEV_1_, (**H**) FVC, or (**I**) post-BD FEV_1_/FVC. LAA, lung attenuation area; %LAA_-950insp_, percentage of lung attenuation area with values less than −950 Hounsfield Units on inspiratory CT scan; BD, bronchodilator.

**Figure 3 biomedicines-10-00786-f003:**
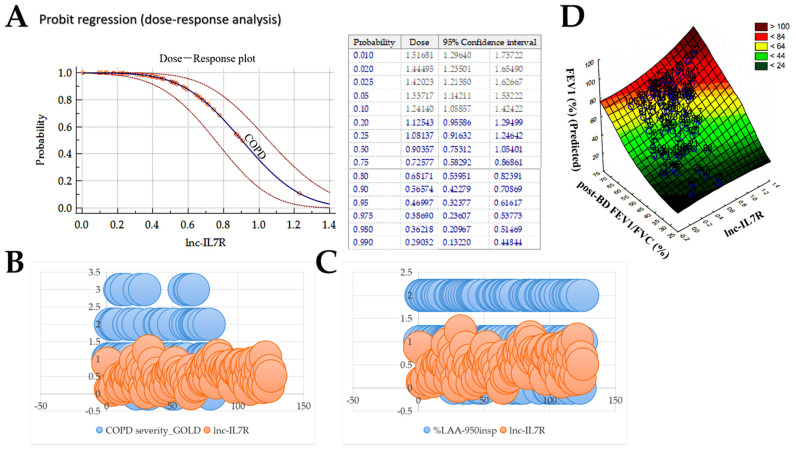
*lnc-IL7R* exhibits strong anti-COPD therapeutic potential. (**A**) Probit regression-based dose–response plot (left panel) and table (right panel) of the effect of *lnc-IL7R* expression on COPD probability. Bubble plots show how *lnc-IL7R* expression stratifies patients according to (**B**) GOLD-defined COPD severity, and (**C**) %LAA_-950insp_-based emphysema severity. (**D**) Surface plot of the correlation between *lnc-IL7R* expression, post-BD FEV_1_/FVC, and predicted FEV_1_.

**Table 1 biomedicines-10-00786-t001:** Baseline characteristics of our COPD cohort (n = 125).

Variables	Patients with COPD (GOLD Stage, n = 125)			
	I (n = 18)	II (n = 58)	III (n = 38)	IV (n = 11)
Age (years)				
Median (IQR)	68.00 (65.25–71.50)	68.50 (62.25–73.00)	70.50 (67.00–77.25)	66.00 (63.00–69.00)
Sex, n (%)				
Male	17 (94.44)	55 (94.83)	31 (81.58)	9 (81.82)
Female	1 (5.56)	3 (5.17)	7 (18.42)	2 (18.18)
BMI, Kg/m^2^				
Median (IQR)	23.90 (21.63–26.29)	24.14 (21.16–26.60)	22.30 (20.00–24.50)	20.60 (19.90–22.98)
Tobacco smoking, n (%)				
Current smoker	5 (27.78)	31 (53.44)	11 (28.95)	2 (18.18)
Ex-smoker	13 (72.22)	23 (39.66)	22 (57.89)	8 (72.73)
Never-smoker	0 (0.00)	4 (6.90)	5 (13.16)	1 (9.09)
Smoking pack-years				
Mean ± SD (Min-Max)	48.89 ± 35.19 (5.00–150.00)	49.02 ± 36.34 (0.00–180.00)	49.30 ± 35.66 (0.00–156.00)	56.73 ± 37.65 (0.00–123.00)
Median (IQR)	42.50 (20.50–60.00)	40.00 (23.00–60.00)	40.00 (25.00–75.00)	46.00 (35.00–85.00)
Pulmonary function indices				
FEV_1_ (L) Median (IQR)	1.90 (1.74–2.11)	1.61 ^b’^ (1.38–1.90	0.99 ^a’b’c’d’^ (0.74–1.12)	0.58 ^a’b’c’d’^ (0.52–0.66)
FEV_1_ % Median (IQR)	84.55 (81.3–86.68)	65.00 ^ab’c^ (57.38–72.00)	39.05 ^a’b’c’d’^ (35.00–45.00)	25.00 ^a’b’c’d’^ (22.10–27.95)
FEV_1_/FVC % Median (IQR)	63.68 (61.25–66.87)	59.25 ^a’b’^ (54.12–65.50)	46.50 ^a’b’c’d’^ (42.11–55.25)	41.41 ^a’b’c’d’^ (30.93–45.67)
Emphysema severity				
Null/Mild (%)	66.67	19.05	0.00	0.00
Moderate (%)	33.33	66.67	69.23	20.00
Severe (%)	0.00	14.28	30.77	80.00
Lnc-IL7R expression				
Median (IQR)	0.88 (0.79–0.94)	0.59 (0.50–0.69)	0.29 (0.19–0.41)	0.18 (0.14–0.43)
COPD exacerbation in previous year				
	1.21 ± 1.03	1.09 ± 1.01	3.59 ± 2.10	3.74 ± 1.35

COPD, chronic obstructive pulmonary disease; GOLD, Global Initiative for Chronic Obstructive Lung Disease; M, male; F, female; FEV1, forced expiratory volume in 1 s; FVC, forced vital capacity; BMI, body mass index; IQR, interquartile range; %LAA_-950insp_, percentages of low attenuation area below—950 Hounsfield units. The values of FEV_1_/FVC % and FEV_1_ % were analyzed by Kruskal-Wallis test and Dunn’s multiple comparison test (^a^
*p* < 0.05, ^a^^’^
*p* < 0.01, compared with non-smoker; ^b’^
*p* < 0.01, compared with smoker; ^c^
*p* < 0.05, ^c’^
*p* < 0.01, compared with COPD patients with GOLD stage I; ^d’^
*p* < 0.01, compared with COPD patients with GOLD stage II.

## Data Availability

The data used in the current study are all contained in the manuscript and may be obtained upon reasonable request from the corresponding author.

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
