# Peer review of "lnc-IL7R Expression Reflects Physiological Pulmonary Function and Its Aberration Is a Putative Indicator of COPD"

_biomedicines, 2022, doi:10.3390/biomedicines10040786_

Round 1

Reviewer 1 Report

In this manuscript, Bamodu et al show the causative association between long non-coding RNA lnc-IL7R and COPD. Despite the methodology is complicated, the ligics is not difficult to understand. The results are presented in a clear way and significant, but higher resolution of each figure is needed.

Reviewer 2 Report

Outstanding job! This is a very nice and well written article with robust data.

I have two comments:

-Please add info about inhaled therapy and make correlations. 

-Please specify whether sample size was calculated. 

Reviewer 3 Report

The manuscript assesses the expression of long non-coding RNA lnc-IL7R in COPD and has found some correlation with lung function and emphysematous phenotype.

I have few major concerns:

  1. The methods are not described adequatly, or in case of lnc-IL7R at all. Lnc-IL7R is not routine hematologic parameter, and methodes used for its determination have to be described in detail (how was blood taken and stored, isolation of RNA, method used for its expression). Also, hematologic parameters and rputine blood workup was mentioned but not at least instruments used were mentioned (and I don't seee any data in manuscript).
  2. Although main focus of the study is differentiation of COVID cases using lnc-IL7R it would be interesting and perhaps give the study more power to also have healthy controld included, and see how lnc-IL7R behaves in that population.
  3. The authors often mention strong correlation for correlation coefficients less than 0.7 , betweern 0,3 and 0,6; which should be either weak or moderate correlation. The authors should amend that throughout the manuscript (in title, abstract, discussion etc.)
  4. In „Clinical relevance“ section point (ii) the authirs state that lnc-IL7R expression in human airway epithelium and blood is clinical relevant. Could they explain whyand how in more detail? Were any studies done on airway epithelium? Does that refer to previous study from the abstract?
  5. Please remove the sentence about previous studies from abstract and concentrate more on current manuscript results and implication. Previous studies should be discussed in Discusion section.
  6. Table 1. is not clear and cluttered, there should be only mean+/- SD or median (IQR) stated, not both. Statistical significance for each parameter shold be clearly marked in the table.
  7. Legend for table one says, Kruskall-Wallis and Dunn's test were used for statistical significance, while methods only mention t-test. Which is correct?

Also, english grammar and language should be checked thoroughout the manuscript.

Round 2

Reviewer 3 Report

The authors sufficiently responded to my earlier comments.